# Non-Targeted Metabolomic Profiling Identifies Metabolites with Potential Antimicrobial Activity from an Anaerobic Bacterium Closely Related to *Terrisporobacter* Species

**DOI:** 10.3390/metabo13020252

**Published:** 2023-02-09

**Authors:** Amila S. N. W. Pahalagedara, Steve Flint, Jon Palmer, Gale Brightwell, Xian Luo, Liang Li, Tanushree B. Gupta

**Affiliations:** 1Food System Integrity Team, AgResearch Ltd., Hopkirk Research Institute, Massey University, Palmerston North 4474, New Zealand; 2School of Food and Advanced Technology, Massey University, Palmerston North 4442, New Zealand; 3Department of Food Science, Aarhus University, Agro Food Park 48, N 8200 Aarhus, Denmark; 4New Zealand Food Safety Science and Research Centre, Massey University, Palmerston North 4474, New Zealand; 5The Metabolomics Innovation Centre and Department of Chemistry, University of Alberta, Edmonton, AB T6N1H1, Canada

**Keywords:** anaerobic bacteria, antimicrobial metabolites, non-targeted metabolomics

## Abstract

This work focused on the metabolomic profiling of the conditioned medium (FS03CM) produced by an anaerobic bacterium closely related to *Terrisporobacter* spp. to identify potential antimicrobial metabolites. The metabolome of the conditioned medium was profiled by two-channel Chemical Isotope Labelling (CIL) LC-MS. The detected metabolites were identified or matched by conducting a library search using different confidence levels. Forty-eight significantly changed metabolites were identified with high confidence after the growth of isolate FS03 in cooked meat glucose starch (CMGS) medium. Some of the secondary metabolites identified with known antimicrobial activities were 4-hydroxyphenyllactate, 3-hydroxyphenylacetic acid, acetic acid, isobutyric acid, valeric acid, and tryptamine. Our findings revealed the presence of different secondary metabolites with previously reported antimicrobial activities and suggested the capability of producing antimicrobial metabolites by the anaerobic bacterium FS03.

## 1. Introduction

Bacteria have been extensively screened for their potential to produce antimicrobial compounds. Secondary metabolites are a good source of antimicrobials and have been implicated in many ways, such as enhancing food quality and safety, with a beneficial effect on human health, and they are effective against multi-drug resistant bacteria. Secondary metabolites such as ribosomally synthesized peptides (RiPPs), non-ribosomal peptides (NRPs), and polyketides (PKs) produced by different bacteria and fungi have been reported to have antimicrobial activity [1,2,3]. Bacteria also produce a broad range of volatile organic compounds as side products of their primary and secondary metabolism [4]. The bacterial organic volatiles mainly include alcohols, aldehydes, ketones, alkenes, terpenes, benzenoids, pyrazines, acids, and esters [5]. Benzenoids have been reported to possess fungicidal, antibacterial, and nematocidal activities [6]. Nitrogen-containing volatile organic product, pyrazines produced by *Bacillus subtilis*, were found to show antibacterial, antifungal, and nematocidal activities [7,8]. *Clostridium difficile* was reported to produce para-cresol through the fermentation of tyrosine, which showed antimicrobial activity against many Gram-negative bacteria [9]. However, bacterial diversity has not been well considered in antimicrobial discoveries, as only the members of a few bacterial groups such as actinomycetes and lactic acid bacteria have been extensively screened for their potential to produce antimicrobials [10]. To date, only a limited number of studies have been conducted to determine the capacity of anaerobic bacteria, particularly within the *Clostridium* genus, to produce antimicrobials through their metabolic pathways [11,12].

Pathogenic bacteria cause a wide variety of infections and poisonings through contaminated foods [13]. Food spoilage microorganisms impact food quality, leading to a reduced shelf life and food loss, particularly during distribution and storage. Certain antimicrobial compounds, such as bacteriocins, phytochemicals, benzoates, sorbates, nitrites, nitrates, and other antimicrobial peptides, are used in controlling foodborne pathogenic and spoilage microorganisms [14,15,16,17,18]. In recent years, natural antimicrobials have gained more attention due to consumer concerns over the adverse health effects of synthetically produced antimicrobials [19]. For instance, although nitrites are permitted in foods as preservatives, there have been concerns about their toxic effects on humans [20,21]. Lactic acid bacteria (LAB) have been used in food bio-preservation from a long time as they may produce a variety of antimicrobial metabolites including bacteriocins that are effective against pathogens [19,22]. However, there is always a need to identify different bacteria that produce more effective antimicrobials and can be used as natural alternatives to synthetic compounds.

*Terrisporobacter* spp. are Gram-positive, spore-forming anaerobic bacteria that were once classified as *Clostridium* spp. [23]. A limited amount of knowledge is available on the different species of *Terrisporobacter* genus, however, the known type strain *T. glycolicus* was first isolated in 1963 from mud and utilizes ethylene glycol as a carbon source [24]. They have also been shown to ferment glucose, fructose, sorbitol, and cellulose and are characterized as fermentative anaerobic bacteria [24]. Furthermore, this species is an emerging pathogen, causing infections in humans [25]. *Terrisporobacter* genus has been shown to play important roles in the organic material degradation of compost and encouraging the humification process [26]. The whole genome sequence of a *T. gylcolicus* strain predicted the presence of a biosynthetic gene cluster encoding an *S*-adenosylmethionine enzyme along with other synthetic genes, which indicated its ability to be an antimicrobial producer. Our previous work demonstrated that the conditioned medium (CM) produced by an anaerobic bacterium (FS03), closely related to *Terrisporobacter* spp., possessed antimicrobial activity against *Bacillus mycoides* ATCC6462, *Bacillus cereus* NZRM5, and *Pseudomonas aeruginosa* ATCC25668 [27]. Conditioned media refer to sterile spent media harvested from the cultured bacterium, which contains metabolites, growth factors, and other extracellular molecules secreted by the bacterial cells.

Therefore, the aim of this study was to evaluate the metabolome of the conditioned medium produced by the isolate FS03 (FS03CM) and identify the metabolites with potential antimicrobial properties. Chemical isotope labelling (CIL) liquid chromatography mass spectrometry (CIL LC-MS) was used in this study, in which heavy isotope-labelled pooled samples can serve as internal standards for each metabolite in individual samples, thus increasing the quantification accuracy in metabolomics. CIL labelling resulted in an increased electrospray ionization (ESI) response and improved LC separation, allowing us to achieve a high coverage in the metabolomic profile analysis of the FS03 cultured growth medium.

## 2. Materials and Methods

### 2.1. Preparation of Conditioned Medium (CM)

Conditioned medium (FS03CM) was prepared from an anaerobic bacterium, FS03, which is closely related to *Terrisporobacter* spp., and had been isolated from soil in a previous study [12] following the method described by Pahalagedara, Jauregui, Maclean, Altermann, Flint, Palmer, Brightwell and Gupta [27]. Briefly, colonies of isolate FS03 grown on sheep blood agar (SBA) were used to inoculate the cooked meat glucose starch (CMGS) medium (45 mL) (Fort Richard Laboratories, New Zealand) supplemented with yeast extract (0.0005%), hemin (0.1%), and vitamin K (1%), and incubated inside a 35 °C anaerobic chamber for 48 h. After the incubation period, the culture was centrifuged at 10,000× *g* for 40 min at 4 °C, and the supernatant was filter-sterilized using a 0.22 µm polyvinylidene fluoride syringe filter (Millipore, Ireland). The sterile conditioned media was aliquoted and stored frozen at −20 °C until use.

### 2.2. Sample Preparation and Chemical Isotope Labelling

Three replicates of FS03CM and CMGS medium, each supplemented with yeast extract (0.0005%), hemin (0.1%) and vitamin K (1%) (10 mL from each of the three replicates), were used for the CIL LC-MS analysis. CMGS medium was included as an uninoculated control in the study for metabolite profile comparison and to identify the compounds generated during the growth of FS03.

A two-channel labelling technique, ^12^C-/^13^C-dansyl chloride (DnsCl) labelling and ^12^C-/^13^C-dimethylaminophenacyl (DmPA) bromide labelling, was carried out for profiling the amine-/phenol and carboxyl- submetabolomes, respectively, in FS03CM and CMGS [28,29]. The total metabolite concentration of each sample was determined by the NovaMT sample normalization kit (Nova Medical Testing, Canada). The total metabolite concentrations of each sample were adjusted to 2 mM prior to labelling. A pooled sample was prepared by mixing equal amounts from each of the concentration-adjusted samples.

The individual samples were labelled with ^12^C_2_-DnsCl or DmPA, and the pooled samples were labelled with ^12^C_2_-/^13^C_2_-DnsCl or DmPA (Nova Medical Testing, Edmonton, AB, Canada) according to the manufacturer’s instructions [30].

### 2.3. LC-MS Analysis

Equal volumes of ^12^C_2_-labelled individual samples and the corresponding ^13^C_2_-labelled pool were mixed before being injecting into the LC-MS. The quality control (QC) sample was prepared by mixing equal volumes of ^12^C_2_-labelled (pooled) and ^13^C_2_-labelled pooled samples. The QC samples were injected after every 3 samples to monitor the LC-MS performance. The retention time calibrant (Nova Medical Testing, Edmonton, AB, Canada) was determined for every 10 samples to monitor the retention time drift and was used for the retention time correction during data processing.

The LC-MS analysis was carried out using an Agilent 1290 LC linked to the Bruker ImpactII quadrupole time-of-flight (Q-TOF) mass spectrometer (Bruker, Billerica, MA, USA) equipped with ESI source. Labelled metabolites were separated using reversed phase liquid chromatography with an Agilent Eclipse Plus C18 column (150 × 2.1 mm, 1.8 µm particle size). The mobile phase A consisted of 0.1% (*v*/*v*) formic acid in water, and the mobile phase B was made up of 0.1% (*v*/*v*) formic acid in acetonitrile. The mobile phase gradient started at 25% B at 0 min, increased to 99% B within 10 min, was held at 99% B for 3 min, decreased to 25% B within 0.1 min, and was held at 25% B until the end of the elution run (16 min). The flow rate was 400 µL/min. The positive ion mode at a spectral acquisition rate of 1 Hz was used to collect all of the MS spectral information within the 220–1000 m/z mass range.

### 2.4. Data Processing

A total of 18 LC-MS data sets (9 LC-MS data, including QCs, from each channel) were uploaded to Bruker Data Analysis software version 4.4 to extract MS spectral peaks. IsoMS Pro software version 1.2.9 was used for the data quality check and data processing, including peak pair picking, peak alignment and missing values filling [31]. For multiple LC-MS runs, the peak pairs that fell in a predetermined mass and retention time window were aligned together to obtain a single file containing peak pair ratio information (relative to the pooled sample) [32].

### 2.5. Metabolite Identification and Statistical Analysis

Metabolite identification was carried out by searching against the CIL library (Tier 1), the Linked Identity (LI) library (Tier 2) and the MCID library (Tier 3). The parameters used are as follows: For the CIL library, the retention time tolerance is 10 s for carboxyl channel and 30 s for the amino/phenol channel; the mass tolerance is 10 ppm. For the LI library, the retention time tolerance is 75 s for the carboxyl channel and 205 s for the amino/phenol channel; the mass tolerance is 10 ppm. For the MCID library, the mass tolerance is 10 ppm. The identified amine/phenol channel dataset and carboxyl channel dataset were merged and subjected to multivariate and univariate analyses using the MetaboAnalyst web tool (https://www.metaboanalyst.ca (accessed on 5 July 2021)).

## 3. Results and Discussion

### 3.1. Multivariate Analysis of FS03CM and CMGS

Multivariate analyses, a principal component analysis (PCA) and a hierarchical cluster analysis were carried out to determine the variability of the samples with regard to the metabolome profiles. In the PCA analysis, the first principal component (PC1) and the second principal component (PC2) accounted for 72.9% and 10% of the overall variance, respectively, (Figure 1a) in the dataset. The FS03CM group was clearly separated from the CMGS group. This indicates that significant changes occurred in the metabolite composition of the CMGS medium during the growth of *Terrisporobacter* spp. FS03. Similar results were obtained through hierarchical cluster analysis, which demonstrated a clear distance between CMGS and FS03CM (Figure 1b).

### 3.2. Univariate Analysis of FS03CM from CMGS

The growth of *Terrisporobacter* spp. FS03 in the CMGS medium considerably changed its metabolite composition after 48 h of incubation compared to that of the control CMGS medium, and this is evident by the results of the multivariate analysis. Therefore, the metabolite composition of F03CM was further evaluated to identify the metabolites that are different from those in the growth medium (CMGS). These significantly discriminating metabolites of FS03CM were produced by the growth of FS03 in the CMGS medium, and they are most likely responsible for the antimicrobial properties of FS03CM reported elsewhere [27]. All of the peak pairs detected in the CMGS and FS03CM groups were used to compute a volcano plot showing the significantly discriminated metabolites in both of the groups (Figure 2). The *x*-axis represents the log_2_ of the magnitude of the fold change (FC) between FS03CM and CMGS, and the *y*-axis represents the negative log_10_ of the *p*-value from the t-test between FS03CM and CMGS. The metabolites with a FC > 2 and *p*-value < 0.05 were considered to be significantly different, and they are denoted in red dots in the volcano plot. A total of 683 metabolites were found to be significantly abundant in FS03CM in comparison to those in CMGS (shown in the right upper quadrant of the volcano plot using red dots). The up-regulated metabolites are most likely produced by the growth of FS03 in CMGS.

### 3.3. Metabolite Identification

Among the 683 significantly abundant metabolites detected in FS03CM, 48 metabolites were positively identified by searching for them in the CIL library (Tier 1) (Table 1). Moreover, a total of 45 peak pairs in FS03CM were putatively identified with a tier 2 identification confidence, which was based on the accurate mass and predicted retention time matches against a linked identity library. As shown in Figure 3, all of the positively identified metabolites were at high levels in FS03CM in comparison to those in CMGS. Most of the identified compounds were linked to protein/amino acid metabolic pathways, indicating the activation of protein metabolism in CMGS by FS03 isolate. This is not surprising because CMGS is an animal protein-rich growth medium, and *Clostridium* spp. and its closely related species are known as protein and amino acid fermentative microorganisms [33]. The identified metabolite list consists of short-chain fatty acids (acetic acid, isobutyric acid and valeric acid), amino acids (glutamine, citrulline, gamma-aminobutyric acid and beta-alanine), amino acid derivatives (tryptomine, succinic semialdehyde and 3-Mmethylhistidine) and metabolites produced from the amino acids (3-hydroxyphenylactic acid, indolelactate).

The tier 1 metabolites were used for the literature research to find information on their antimicrobial potential. The outcome of the literature research showed that 4-hydroxyphenyllactate, 3-hydroxyphenylacetic acid, acetic acid, isobutyric acid, valeric acid and tryptamine, which were positively identified from FS03CM, had previously been reported to possess antimicrobial activity against some microorganisms [34,35,36,37]. Hydroxyphenyllactic acid or 4-hydroxyphenyllactate (4-HPLA) is a metabolite that is produced through the breakdown of the amino acid tyrosine [38] and is found to be produced by bacteria and has been reported to possess an antifungal property [34].

Another metabolite produced by FS03 using CMGS as the growth medium was 3-hydroxyphenylacetic acid (3-HPAA). During the metabolism of tyrosine, 3-HPAA is a metabolite that is also produced [39]. This compound has been reported to have antimicrobial activity against *Pseudomonas aeruginosa* by affecting multiple bacterial processes such as DNA replication and repair, RNA modifications, cell envelope and oxidative stress [35]. As there is no information on its antimicrobial efficacy against other bacteria, it will be worth investigating its activity against a wide range of bacteria to determine its potential antimicrobial applications.

Acetic acid is a monocarboxylic acid, and it has been applied to medicine and the food industry. *Clostridium* spp. have been reported to produce acetic acid through the anaerobic fermentation of glucose and cellulose [40]. The current study reported on the production of acetic acid by FS03 in CMGS medium that contains glucose. Acetic acid has a long history as an effective antiseptic agent [36]. Acetic acid is used in the food industry as a natural preservative and acidulant and has been recognized as a GRAS (generally recognized as safe) additive [41]. Acetic acid has been used in form of spray or dips to reduce the microbiological contamination of meat carcasses [42].

Isobutyric acid/isobutanoic acid is a carboxylic acid and a branched short-chain fatty acid. It is produced by the fermentation of branched amino acids resulting from undigested proteins in the colon by the gut microbiota in humans [43]. *Clostridium uticellarii* have been reported to produce isobutyric acid from methanol [44]. The present study reported the production of isobutyric acid by FS03, which is closely related to *Terrisporobacter* spp., in an animal protein-rich CMGS medium. This compound has been applied to the food industry as a flavoring agent and is permitted for direct addition to food for human consumption by the U.S. Food and Drug Administration (FDA) [45]. A previous study investigated the antibacterial activity of short-chain fatty acids (SCFAs), including isobutyric acid, against several oral microorganisms and demonstrated that isobutyric acid was active against all test bacteria at concentrations ranging from 1.4 mg/mL to over 2.5 mg/mL [37].

Valeric acid/pentanoic acid is an alkyl carboxylic acid detected in human feces and known to be produced by gut microbiota, including some *Clostridium* species [46]. This metabolite is synthesized through the microbial fermentation of the amino acids, such as proline and hydroxyproline [47,48]. The current study suggests the production of valeric acid by isolate FS03 in CMGS medium. Kovanda, et al. [49] investigated the antimicrobial activity of seven organic acids and their derivatives, including valeric acid, and reported that valeric acid was active against both Gram-positive and Gram-negative bacteria with minimum inhibitory concentrations (MICs) ranging between 0.5–3.2 mg/mL. Another metabolite significantly abundant in FS03CM compared to CMGS was tryptamine. It has been reported that the gut microorganisms that express tryptophan decarboxylase can produce tryptamine from tryptophan [50]. This group of compounds possesses mainly hallucinogenic effects interacting with the neurotransmitter system in humans [51]. Tryptamine and its derivatives have also been reported to have antimicrobial activities against some bacteria and yeast [52,53].

This work found that the anaerobic bacterium FS03 produced several short-chain fatty acids, including acetic acid, isobutyric acid and valeric acid, as a result of its metabolic activities during the growth in cooked meat glucose starch medium (CMGS). These SCFAs are some of the end products of the fermentation of dietary proteins and fibers by anaerobic intestinal microbiota in the human colon [47]. The SCFAs in the colon are believed to play a vital role in human health, including the inhibition of harmful bacteria and the regulation of inflammation, intestinal barrier functions and oxidative stress [54,55]. Previous studies have shown that anaerobic bacteria in the gut provide a protective effect on the liver by decreasing the inflammatory responses, oxidative damage and lipogenesis in liver tissues through the production of SCFAs [56]. Moreover, probiotic bacteria such as *Lactobacilli* spp. and *Bifidobacterium* spp. have been reported to produce SCFAs [57,58]. It is proposed that SCFAs, together with other compounds such as hydrogen peroxide, bacteriocins and bacteriocin-like inhibitory substances (BLIS) produced by probiotic bacteria, are responsible for their protective effect against harmful bacteria [59]. The metabolome of FS03CM, together with its reported antimicrobial potential, suggests that FS03 possesses some characteristics of a potential probiotic bacterium. However, more research needs to be conducted to prove this potential.

The growing trend to use antimicrobials from natural and renewable sources, such as microorganisms, particularly in food preservation due to consumer concerns and the impact on environment [60], has led to the quest of identifying new natural antimicrobial sources. The current study highlights the significance of further investigating the use of anaerobic bacteria, such as the FS03 isolate, as a potential source to produce antimicrobial metabolites, which may be applied to food preservation/medicine.

In this study, a CIL-LC-MS (chemical isotope labelling-LC-MS) technique was used for the in-depth metabolome analysis with good quantification accuracy [30]. After chemical isotope labeling, the metabolites are easier to be ionized during the ESI process, thereby enhancing the metabolite signal intensity by 10–1000 fold, making the detection of low-abundant metabolites easier. The present work included a two-channel labelling techniques, ^12^C-/^13^C-dansyl chlorite and ^12^C-/^13^C-dimethylaminophenacyl (DmPA) bromide, targeting the metabolites containing amine/phenol and carboxyl submetabolome analysis. The use of two channels certainly increased the metabolome coverage of FS03CM. However, one of the limitations of this study was not targeting metabolites with other functional groups such as hydroxyls and carbonyls. Therefore, this study might have missed the detection of some metabolites with those functional groups. Additionally, the tier 1 identification rate also needs to be improved to cover more potential antimicrobial activity-related metabolites.

## 4. Conclusions

The present study showed significant changes in the metabolite composition of CMGS after the growth of isolate FS03, which is closely related to *Terrisporobacter* spp., using both univariate and multivariate data analyses. The metabolite identification with the highest confidence level (tier 1) confirmed 48 metabolites produced by FS03 in the CMGS medium. Various metabolites from FS03CM, including 4-hydroxyphenyllactate, 3-hydroxyphenylacetic acid, acetic acid, isobutyric acid, valeric acid, and tryptamine, with previously reported antimicrobial activity against certain bacteria, were positively identified. Overall, this study revealed that an anaerobic bacterium closely related to *Terrisporobacter* spp. could produce many metabolites with potential antimicrobial activities and many uncharacterized metabolites during its growth in the CMGS medium. Further research should focus on assessing the antimicrobial potential (both the previously known and unknown ones) of identified metabolites against bacteria associated with food safety, quality and human health to understand their future applications. Future research should also focus on enhancing the isolation and purification of key metabolites by utilizing targeted metabolomics.

## Figures and Tables

**Figure 1 metabolites-13-00252-f001:**
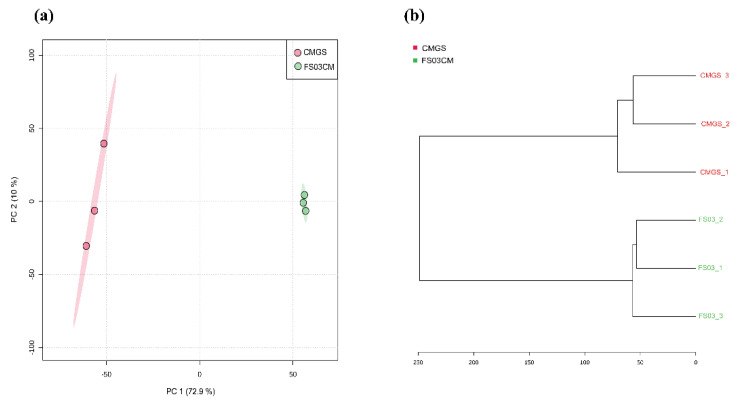
Two-dimensional PCA score plot and hierarchical cluster analysis dendrogram of metabolomics data obtained for FS03CM and CMGS. PCA score plot shows the differences in metabolome profiles of FS03CM and CMGS (**a**). Dendrogram shows the relationship between FS03CM and CMGS medium based on their metabolomic profiles (**b**). Dendrogram was constructed using ward clustering on the closest Euclidean distances between samples. Samples are color coded as indicated in the figure legend. Numbers represent sample IDs.

**Figure 2 metabolites-13-00252-f002:**
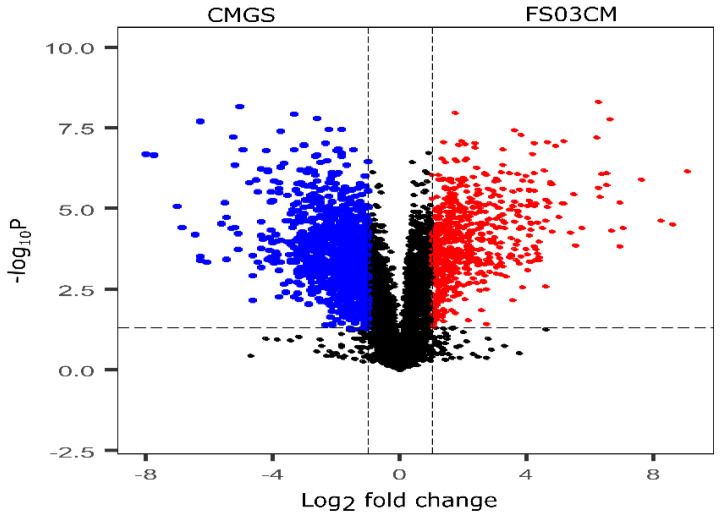
Volcano plot shows the relative abundance of metabolites in CMGS and FS03CM groups. It was constructed by combining the statistical t-test [-log_10_(*p*-value)] and the magnitude of the fold change [log_2_(FC)]. Red dots represent the metabolites that are significantly abundant in FS03CM (*p*-value < 0.05 and FC > 2). Blue points represent the metabolites that are significantly abundant in CMGS (*p*-value < 0.05 and FC > 2. Black dots represent the rest of the metabolites that were detected in both samples. CMGS (cooked meat glucose starch media, FS03CM (conditioned medium prepared from FS03).

**Figure 3 metabolites-13-00252-f003:**
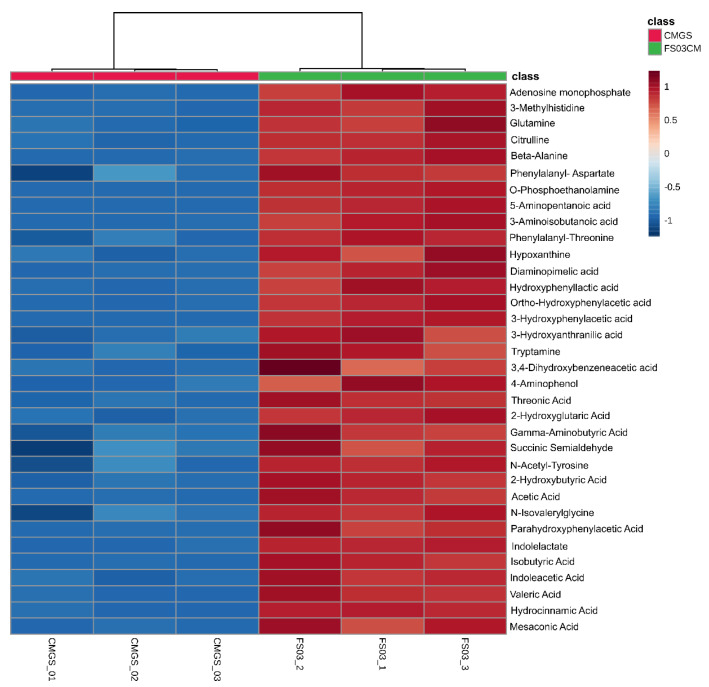
Peak pair ratio of positively identified metabolites (with tier 1 identification confidence). Heatmap shows positively identified metabolites with their abundance in FS03CM and CMGS groups. Rows represent positively identified metabolites, whereas columns represent replicates (n = 3). The levels of metabolites are shown using a pseudo-color scale (from −1.0 to 1.0), with red representing high intensity levels and blue representing low intensity levels. CMGS (Cooked meat glucose starch medium); FS03CM-conditioned medium prepared from FS03.

**Table 1 metabolites-13-00252-t001:** Positively identified (tier 1) significantly abundant metabolites in FS03CM.

External Identifier	Compound Name	Neutral Mass (Da)	Normalized RT (s)
C02043	Indolelactate	205.0722	320.7
C05629	Hydrocinnamic Acid	150.0686	426.5
C05145	3-Aminoisobutanoic Acid	103.0625	208.7
C05984	2-Hydroxyglutaric Acid	148.0361	340.7
C01620	Threonic Acid	136.0387	130
C00327	Citrulline	175.0963	222.1
C00954	Indoleacetic Acid	175.0631	364.6
HMDB0029005	Phenylalanyl-Threonine	266.1259	530.6
C00398	Tryptamine	160.1009	1070.7
C00346	O-Phosphoethanolamine	141.0197	121.4
C00632	3-Hydroxyanthranilic acid	153.0422	1070.4
C00334	Gamma-Aminobutyric Acid	103.0622	176.2
C00803	Valeric Acid	102.0689	404.9
C05852	3-Hydroxyphenylacetic acid	152.0483	990.4
C00431	5-Aminopentanoic acid	117.0803	513.2
C05852	Ortho-Hydroxyphenylacetic acid	152.0471	980.7
C00099	Beta-Alanine	89.0481	430.3
C02632	Isobutyric Acid	88.053	352.6
C00232	Succinic Semialdehyde	102.0319	226.9
HMDB0000479	3-Methylhistidine	169.0846	121.9
C00020	Adenosine monophosphate	347.0643	110.2
C00064	Glutamine	146.0689	216.8
C00262	Hypoxanthine	136.0392	545.6
C00033	Acetic Acid	60.021	247.6
C03672	Hydroxyphenyllactici acid	182.0573	837.3
C00666	Diaminopimelic acid	190.094	740
HMDB0000678	N-Isovaleroylglycine	159.0887	260.5
C02372	4-Aminophenol	109.0511	1473.3
HMDB0028991	Phenylalanyl-Aspartate	280.1052	493.6
C01732	Mesaconic Acid	130.0261	441.4
C00642	Parahydroxyphenylacetic Acid	152.0478	291.7
HMDB0000866	N-Acetyl-Tyrosine	223.083	227.3
C01161	3,4-Dihydroxybenzeneacetic acid	168.0427	1422.5
C2136	2-Hydroxybutyric Acid	104.0472	243

External identifier—KEGG/HMDB entry of the identified compound; neutral mass (Da)—the neutral monoisotope mass of the metabolite (i.e., labelled mass—the mass of the labelling group); normalized RT (s)—the corrected retention time of the peak pair with universal RT Calibrant data.

## Data Availability

Data will be made available upon request from the corresponding author. The data are not publicly available due to privacy.

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
