# Peer review of "Non-Targeted Metabolomic Profiling Identifies Metabolites with Potential Antimicrobial Activity from an Anaerobic Bacterium Closely Related to Terrisporobacter Species"

_metabolites, 2023, doi:10.3390/metabo13020252_

Round 1

Reviewer 1 Report

The present manuscript investigates metabolic activity and alteration of cooked meat glucose starch during the growth of Terrisporobacter-like bacterium. The research is interesting, even-though it looks like a small part of a previous research that the authors conducted. The introduction section has no useful information on the subject. Lines 36-38, please be specific. Please provide information on the nature of Terrisporobacter and metabolic activity reported from similar studies. The identification of metabolites was the principal aim of the study, however the results do not present the metabolites in a separate analytical paragraph and the discussion to justify the results is practically absent. The list of references is limited and the authors reference their own work.

Author Response

Reviewer 1

We would like to express our sincere gratitude for taking the time to review our manuscript and provide valuable feedback to help improve the manuscript.

Comment:

The introduction section has no useful information on the subject. Lines 36-38, please be specific. Please provide information on the nature of Terrisporobacter and metabolic activity reported from similar studies.

Response:

We have now enriched the introduction section by providing information on various types of antimicrobial metabolites reported to be produced by bacteria, their applications, and the need for further research on identifying bacteria with antimicrobial properties, the characteristics of Terrisporobacter species, and other available information has also been discussed (Lines 33 – 79).  Changes are marked in red.

Comment:

The identification of metabolites was the principal aim of the study, however the results do not present the metabolites in a separate analytical paragraph and the discussion to justify the results is practically absent.

Response:

A new paragraph and Table 1 have been added to section 3.3 to present the significantly abundant metabolites identified from FS03CM and to justify the results (Line 272 – 348). Text marked in red.

Comment:

The list of references is limited, and the authors reference their own work.

Response:

Now we have added information to the manuscript from 61 references (Line 496 – 643). Our previous papers have only been cited to support the methods used in this study and to justify the current work.

Reviewer 2 Report

Authors have done a very good work. However, following things may be addressed to improve the MS further:

-Authors stated that “. Our findings revealed the presence of different secondary metabolites with previously reported antimicrobial activities in FS03CM, which suggests the capability of producing anti- 26 microbial metabolites by the anaerobic bacterium FS03.” However, it would be better to as add one or two sentences about application of these metabolites for therapeutic purposes to improve the importance of the current work. Same should be done in the conclusion section.

-How will go for high-throughput isolation and purification of these secondary metabolites? This is essential to have an industrial microbiological application of the current work.

-In many cases as per my experience, MS results could not be reproduced for metabolite detection and their levels. How will you assure that reproducibility? Could you check the results for twice or thrice and found same results?

Author Response

Reviewer 2

We would like to express our sincere gratitude for taking the time to review our manuscript and provide valuable feedback help improve the manuscript.

Comment:

Authors stated that “. Our findings revealed the presence of different secondary metabolites with previously reported antimicrobial activities in FS03CM, which suggests the capability of producing anti- 26 microbial metabolites by the anaerobic bacterium FS03.” However, it would be better to as add one or two sentences about application of these metabolites for therapeutic purposes to improve the importance of the current work. Same should be done in the conclusion section.

Response:

New paragraph has been added to describe the applications of antimicrobial compounds (particularly in food products) and the need for further research on identifying bacteria with antimicrobial properties, (Line 52-64 and 421-443). Changes have been made throughout the document. Changes are marked in red  

Comment:

How will go for high-throughput isolation and purification of these secondary metabolites? This is essential to have an industrial microbiological application of the current work.

Response:

This is a very interesting engineering topic, although a little bit beyond of our manuscript scope, we will try our best to answer your questions. In fermentation engineering, there are several approaches to enrich and separate secondary metabolites. We list some of them.

  1. Macroporous adsorbents

J Chem Technol Biotechnol  2018;93: 3176–3184

The authors used macroporous adsorbents to separate phenazine-1-carboxylic acid from Pseudomonas chlororaphis GP72.

  1. Membrane separation

Ind. Eng. Chem. Res. 2017, 56, 29, 8301–8310

The authors used integrated membrane separation to separate lactic acid.

  1. Solvent extraction

Eng. Life Sci.2014,14, 108–117

This is a good review paper of salting-out extraction.

There are some other approaches (e.g., distillation, crystallization). Sometimes, combination of several different approaches is employed for separation of secondary metabolites. 

Comment:

In many cases as per my experience, MS results could not be reproduced for metabolite detection and their levels. How will you assure that reproducibility? Could you check the results for twice or thrice and found same results?Now we have added information to the manuscript from 61 references (Line 444 – 594). Our previous papers have only been cited to support the methods used in this study and to justify the current work.

We agree with you. In conventional LC-MS based metabolomics, MS signal intensity drift is always observed from sample to sample. And low abundant metabolites can’t be detected routinely. However, in this study, we employed Chemical Isotope Labeling (CIL) LC-MS for metabolomic profiling. After chemical isotope labeling, the metabolites are easier to be ionized during ESI process. Therefore, the metabolite signal intensity can be enhanced by 10-1000 folds, making the detection of low abundant metabolites more routinely. In CIL LC-MS, 13C-labeled pool serves as internal standards, and the MS intensity drift can be corrected by 13C-labeled pool. Thus, the reproducibility of CIL LC-MS based metabolomics is much improved. This can be demonstrated by tightly clustered QCs in PCA plot (Line 196-210).

Please see the reference below.

Zhihua Li*, Ling Dong, Chi Zhao, Fengju Zhang, Shuang Zhao, Jingjing Zhan, Jia Li and Liang Li*, 2022, “Development of a high-coverage quantitative metabolome analysis method using 4-channel chemical isotope labeling LC-MS for analyzing high-salt fermented food”, Journal of Agricultural and Food Chemistry, 70, 28, 8827–8837.

Reviewer 3 Report

Major revision

In the manuscript entitled “Non-targeted metabolomic profiling identifies metabolites with potential antimicrobial activity from an anaerobic bacterium closely related to Terrisporobacter species” the authors investigated the antimicrobial metabolites produced by Terrisporobacter species through metabolomic analysis.

Overall, the work is well done, carefully thought out, and performed with a consistent statistical analysis. This is an interesting topic, and it is an area that really needs our attention. However, there are still some areas of the article that need to be enriched. After the revision, the article could be considered for publication in the prestigious Metabolites Journal.

1.      The introduction encompassed the capability of bacteria to produce antimicrobial compounds. Since the literature of this part is scarce, I suggest enriching the literature with the pathway involved and the applications of antimicrobial activity reflecting on the quality and nutritional value of foods (an important theme that has raised much attention). Perhaps, to enrich the scientific relevance, these articles are helpful to be cited:

-        Antimicrobial peptide antibiotics inhibit aerobic denitrification via affecting electron transportation and remolding carbon metabolism.

-        Effects of Grape Pomace Polyphenols and In Vitro Gastrointestinal Digestion on Antimicrobial Activity: Recovery of Bioactive Compounds.

-        Application of lactic acid bacteria for the biopreservation of meat products: A systematic review.

2.      The paragraph 3.3. needs more discussion. Perhaps, to enrich the scientific relevance, these articles are helpful to be cited:

-        Evaluation of the Effects of the Tritordeum-Based Diet Compared to the Low-FODMAPs Diet on the Fecal Metabolome of IBS-D Patients: A Preliminary Investigation.

-        Microbial ecology of anaerobic digesters: the key players of anaerobiosis.

-        A Low Glycemic Index Mediterranean Diet Combined with Aerobic Physical Activity Rearranges the Gut Microbiota Signature in NAFLD Patients.

Other specific comments:

- Please carefully check grammatically and spelling throughout the manuscript.

-English language and style are fine/minor spell check required

Author Response

Reviewer 3

We would like to express our sincere gratitude for taking the time to review our manuscript and provide valuable feedback to help improve our manuscript.

Comment:

The introduction encompassed the capability of bacteria to produce antimicrobial compounds. Since the literature of this part is scarce, I suggest enriching the literature with the pathway involved and the applications of antimicrobial activity reflecting on the quality and nutritional value of foods (an important theme that has raised much attention). 

Response:

We have now enriched the introduction section by providing information on various types of antimicrobial metabolites reported to be produced by bacteria, their applications, and the need for further research on identifying different bacteria with antimicrobial properties. We have tried to improve the introduction as much as possible, according to your valuable suggestions (See introduction with red marked paragraphs and sentences). 

Comment:

The paragraph 3.3. needs more discussion. Perhaps, to enrich the scientific relevance, these articles are helpful to be cited

Response:

A new paragraph has been added to section 3.3 to present the scientific relevance of our results (Line 374 – 394). More information has also been added to section 3.3 to show the pathways and known applications of identified metabolites with reported antimicrobial activities. Please see changes marked in red in section 3.3

Comment:

Please carefully check grammatically and spelling throughout the manuscript.

English language and style are fine/minor spell check required.

The manuscript was checked for grammar and spelling mistakes.

Round 2

Reviewer 1 Report

Manuscript could be accepted for publication 

Reviewer 3 Report

Thanks for the revised manuscript. After the suggested improvement, the article can be considered for publication.